# The structural dynamics of macropinosome formation and PI3-kinase-mediated sealing revealed by lattice light sheet microscopy

Shayne E. Quinn [1,2], Lu Huang[3,4], Jason G. Kerkvliet[4,5], Joel A. Swanson [6], Steve Smith[1,2], Adam D. Hoppe [4,5], Robert B. Anderson[1,2✉], Natalie W. Thiex[3,4✉] & Brandon L. Scott [1,2✉]

Macropinosomes are formed by shaping actin-rich plasma membrane ruffles into large intracellular organelles in a phosphatidylinositol 3-kinase (PI3K)-coordinated manner. Here, we utilize lattice lightsheet microscopy and image visualization methods to map the three-dimensional structure and dynamics of macropinosome formation relative to PI3K activity. We show that multiple ruffling morphologies produce macropinosomes and that the majority form through collisions of adjacent PI3K-rich ruffles. By combining multiple volumetric representations of the plasma membrane structure and PI3K products, we show that PI3K activity begins early throughout the entire ruffle volume and continues to increase until peak activity concentrates at the base of the ruffle after the macropinosome closes. Additionally, areas of the plasma membrane rich in ruffling had increased PI3K activity and produced many macropinosomes of various sizes. Pharmacologic inhibition of PI3K activity had little effect on the rate and morphology of membrane ruffling, demonstrating that early production of 3′-phosphoinositides within ruffles plays a minor role in regulating their morphology. However, 3′-phosphoinositides are critical for the fusogenic activity that seals ruffles into macropinosomes. Taken together, these data indicate that local PI3K activity is amplified in ruffles and serves as a priming mechanism for closure and sealing of ruffles into macropinosomes.

[1] Nanoscience and Nanoengineering, South Dakota School of Mines and Technology (South Dakota Mines), Rapid City, SD, USA. [2] BioSNTR, South Dakota Mines, Rapid City, SD, USA. [3] Department of Biology and Microbiology, South Dakota State University (SDSU), Brookings, SD, USA. [4] BioSNTR, SDSU, Brookings, SD, USA. [5] Department of Chemistry and Biochemistry, SDSU, Brookings, SD, USA. [6] Department of Microbiology and Immunology, University of Michigan, Ann Arbor, MI, USA. ✉email: robert.anderson@sdsmt.edu; natalie.thiex@sdstate.edu; brandon.scott@sdsmt.edu

Macropinocytosis, or "cell drinking," is a form of clathrin-independent endocytosis that results in the non-specific uptake of large volumes of extracellular fluid and solutes. This central macrophage function enables immune surveillance, clearing of debris, and sampling of the local environment for the presence of pathogen- or damage-associated molecular patterns, cytokines, growth factors, nutrients, and other soluble cues[1–6]. Macropinosomes also serve as platforms to integrate this diverse information and to activate a variety of signaling pathways[7–10]. The major macrophage growth factor, colony-stimulating factor-1 (CSF-1), stimulates macropinocytosis and contributes to ligand-dependent modulation of CSF-1 receptor signaling[9]. Additionally, cytokines such as C-X-C chemokine motif ligand-12 and the bacterial cell wall component lipopolysaccharide (LPS) acutely stimulate macropinocytosis[5,11,12].

Construction of a macropinosome proceeds through autonomous, ligand-independent plasma membrane extensions known as ruffles, which are driven by actin polymerization and require the phosphorylation and dephosphorylation of different signaling phospholipids[2,13]. In the closely related process of solid particle uptake, the shape of the particle is used to template the phagosome structure[14,15]. In contrast, ruffles that fuse into macropinosomes do not have a structural framework to use as a template resulting in various potential closing mechanisms including 'purse string' closure of circular dorsal ruffles[13], closure at the distal tips of ruffles[7], and more recently described closure following actin tentpole crossing[12]. Regardless, the result is an internal organelle derived from the plasma membrane that is filled with extracellular fluid[16].

The dynamic lipid microenvironment impacts the localization of downstream effector molecules that drives actin polymerization and ruffle growth into macropinosomes[17]. The production of 3′ phosphoinositides (PIs) by PI 3-kinase (PI3K) is required to generate isolated patches of phosphatidylinositol 3,4,5,triphosphate (PIP$_3$) on the plasma membrane[18,19], and the sequential breakdown of PIP$_3$ into PI(3,4)P$_2$ and ultimately PI is necessary for successful macropinosome formation[20]. It is only in the cellular slime mold *Dictyostelium* that the signal coordination throughout the three-dimensional (3D) ruffle volume during macropinocytosis has been well described[18], and there is still more to learn about how these events are spatially coordinated in metazoan cells[21]. The precise membrane dynamics of macropinocytosis and the spatial coordination of PI3K in forming ruffles remains unclear because of the low spatial and temporal resolution of other microscopy approaches. Previously, widefield ratiometric imaging has shown that PIP$_3$ concentration peaks after ruffle circularization[17,22,23], and PI3K inhibitors, including LY294002, have demonstrated that PI3K activity is only required for macropinosome closure but does not inhibit ruffling[17]. Recently, high-resolution imaging of macropinocytosis in a macrophage-like cell line (RAW 264.7) indicated that the prior models of macropinocytosis may be more diverse than previously thought[12]. With the variety of different macrophage-like cell lines available, all of which perform constitutive macropinocytosis while displaying different phenotypes and protein expressions[24], it is more important than ever to expand the 3D understanding of membrane dynamics and protein localization during macropinocytosis.

Here we employ the powerful 3D imaging capabilities of lattice light sheet microscopy[25] (LLSM) and volumetric image analysis to create high-resolution movies of plasma membrane dynamics and PI3K activity during ruffling and macropinocytosis. The images and movies we present advance our understanding of the spatial dynamics of membrane ruffling, the morphologies that lead to macropinosomes, the spatial distribution of PI3K activity

during macropinocytosis, and finally acknowledge the structural variability among different cell lines during macropinocytosis in fetal liver macrophages (FLMs), bone marrow-derived macrophages (BMDMs), and RAW 264.7 macrophages. Our results show that the majority of macropinosomes form by non-specific collisions of adjacent PI3K-rich ruffles. We show that PI3K activity is present at the earliest stages of ruffle extensions and is highly localized to the bottom of ruffles after the membrane has closed into a macropinosome. Finally, we modulate the rate of macropinocytosis using stimulation and pharmacological inhibition to demonstrate that the ruffle morphology is unaffected, but PI3K activity is required to prime ruffle membranes for sealing into macropinosomes.

## Results

**LLSM allows volumetric visualization of plasma membrane movements relative to PIP$_3$ and PI(3,4)P$_2$ distribution during macropinocytosis.** Our first objective was to capture the 3D structure of the plasma membrane relative to PI3K activity during macropinosome formation. LLSM imaging was performed on FLMs stably expressing the fluorescent proteins mNeonGreen localized to the plasma membrane via the lipidation signal sequence from Lck (Mem-mNG) and the pleckstrin homology domain of Akt fused to mScarlet-I (AktPH-mSc). The AktPH probe recognizes PIP$_3$ and PI(3,4)P$_2$ with similar affinity and has been used extensively to characterize PI3K activity during macropinocytosis[8]. Using LLSM imaging in conjunction with the molecular visualization software, ChimeraX[26], we represent the data using the following renderings: isosurface, surface mesh, volumetric maximum intensity, and orthogonal planes. An isosurface is a 3D contour map that represents points in volume space as constant values at a given intensity threshold causing all pixels above the threshold to be shown as a solid-colored voxel (value of 1), while all voxels below the threshold are transparent/hidden (value of 0) resulting in a crisp surface depiction of the membrane probe (Fig. 1a). These images were of sufficient resolution that the detailed structure of ruffles and forming macropinosomes could be observed in living cells (Fig. 1a), similar to scanning electron microscopic imaging of BMDMs (Fig. 1b). The isosurface is rendered with ambient occlusions meaning that any internal information is hidden from view. To visualize the recruitment of AktPH-mSc relative to the membrane, we first used volumetric intensity renderings that display the most intense color value underlying the pixel along the line of sight (ChimeraX User Guide), which provides a visually intuitive method of displaying 3D fluorescent intensities (Fig. 1c). The resulting volume is similar to a maximum intensity projection; however, the projection dimension is dependent on the orientation, i.e., top–down would yield an *xy*-maximum intensity projection collapsed in *z* and a side view would provide the corresponding *xz*/*yz* maximum intensity collapsed in *y*/*x*, respectively. As can be seen in the volume renderings, Mem-mNG persisted on newly formed intracellular vesicles derived from the plasma membrane (Fig. 1c). Moreover, we observed membrane movements throughout the entire formation and early trafficking of macropinosomes, as well as the recruitment of AktPH-mSc to forming macropinosomes (Fig. 1c (arrow) and Supplementary Movie 1). It is difficult to perceive depth in the still-frame volumetric renderings but orthogonal plane slices (orthoplanes) in *xy*, *yz*, and *xz* (~0.1 μm thick) showed that AktPH-mSc was enriched in ruffles to varying degrees and intensely labeled circular structures found near the base and sides of ruffles (Fig. 1d and Supplementary Movie 2). While orthoplanes are effective for examining the two-dimensional (2D) relationships between the fluorescent signals, they can also

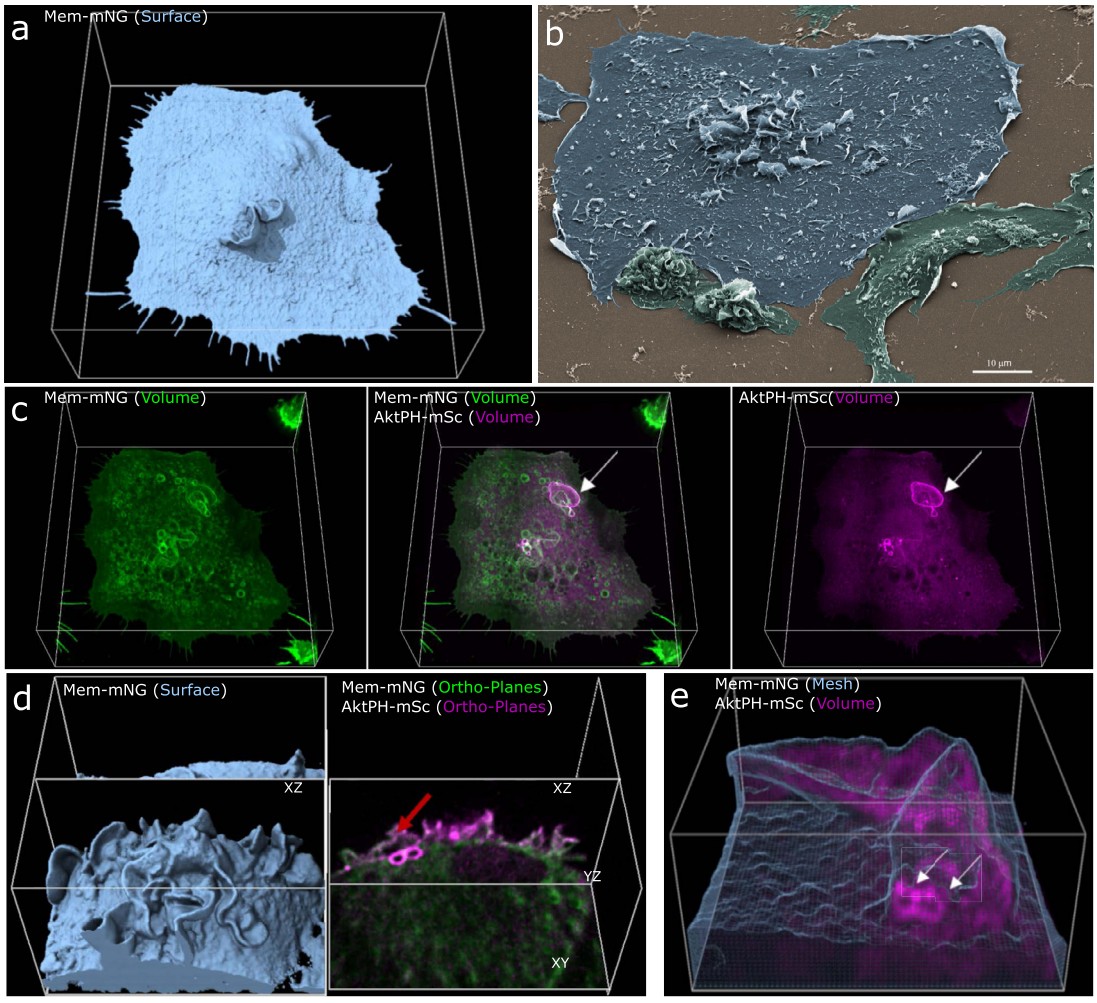

**Fig. 1 3D visualization of macrophages allows insight into membrane structure and phosphatidylinositol dynamics during macropinocytosis. a** Isosurfaces show the plasma membrane of a live cell that is actively macropinocytosing (region 68 × 72 × 25 μm *x, y, z*). **b** SEM image of a macrophage acutely stimulated with CSF-1 shows representative high-resolution fixed cells (*n* = 10 micrographs, scale bar is 10 μm). **c** Volumetric intensities show specific local fluorescence (left→right) volumetric membrane (green), dual volumetric membrane and AktPH-mSc, volumetric AktPH-mSc (magenta). Volumetric renderings provide a method to visualize the transient fluorescent intensities throughout the volume of the cell (region 68 × 72 × 25 μm). **d** Combinations of visualization techniques such as isosurface (left) displayed alongside orthogonal planes (right) further clarify how each plane is chosen to show internal intensities (region 29 × 30 × 19 μm). **e** Mesh rendering of the Mem-mNG probe along with volumetric AktPH-mSc provides a representation of the plasma membrane structure as well as underlying fluorescence. The white arrows indicate the post closure recruitment of AktPH-mSc (region 13 × 14 × 7 μm). Different rendering methods provide insight into cellular characteristics, such as structure, depth, and fluorescent intensity and provide a foundation for visualizing localization of AktPH-mSc to the constantly changing plasma membrane during macropinocytosis.

produce incomplete or distorted perspectives that are resolved by viewing the full volumetric data (Supplementary Fig. 1), such as when a macropinosome appears closed vs open. To overcome these limitations, we implemented a mesh derived from the Mem-mNG isosurface with transparent polygonal faces (only displaying edges and vertices) that enables visualizing the underlying volumetric AktPH-mSc signals, while maintaining the structural framing needed to resolve plasma membrane rearrangements (Fig. 1e).

For visualizing a macropinosome sealing event, we began by imaging a representative cell for these data that retained the canonical morphology like that of BMDMs with a profile view representing a "fried egg" ranging from ~1 μm thick at the outer edge and up to ~6 μm thick near the center. Next, an XY-MIP was used to quickly find potential closures using the AktPH-mSc signal as an indicator. The identified coordinate was then translated to the corresponding isosurface, and the timepoint at

which the isosurface was fully closed was considered as a nascent macropinosome. These formations were examined using the full volumetric intensity projections for further analysis. Together, combinations of these visualization techniques were applied to 122 macropinosome formations and enabled correlating the location and timing of PI3K activity to the membrane extension, curvature, and fusion of macropinosomes with unprecedented spatial and temporal resolution.

**AktPH-mSc is recruited early during ruffle expansion and peaks at the base of ruffles after macropinosome sealing.** In prior analysis of macropinocytosis using microscopic methods with low axial resolution, AktPH recruitment was identified as ruffles that transitioned into closed circular ruffles and nascent macropinosomes[8]. Here the enhanced *z*-axis resolution and detection sensitivity of LLSM enabled visualizing the dynamic recruitment of AktPH-mSc to ruffles as they began to protrude

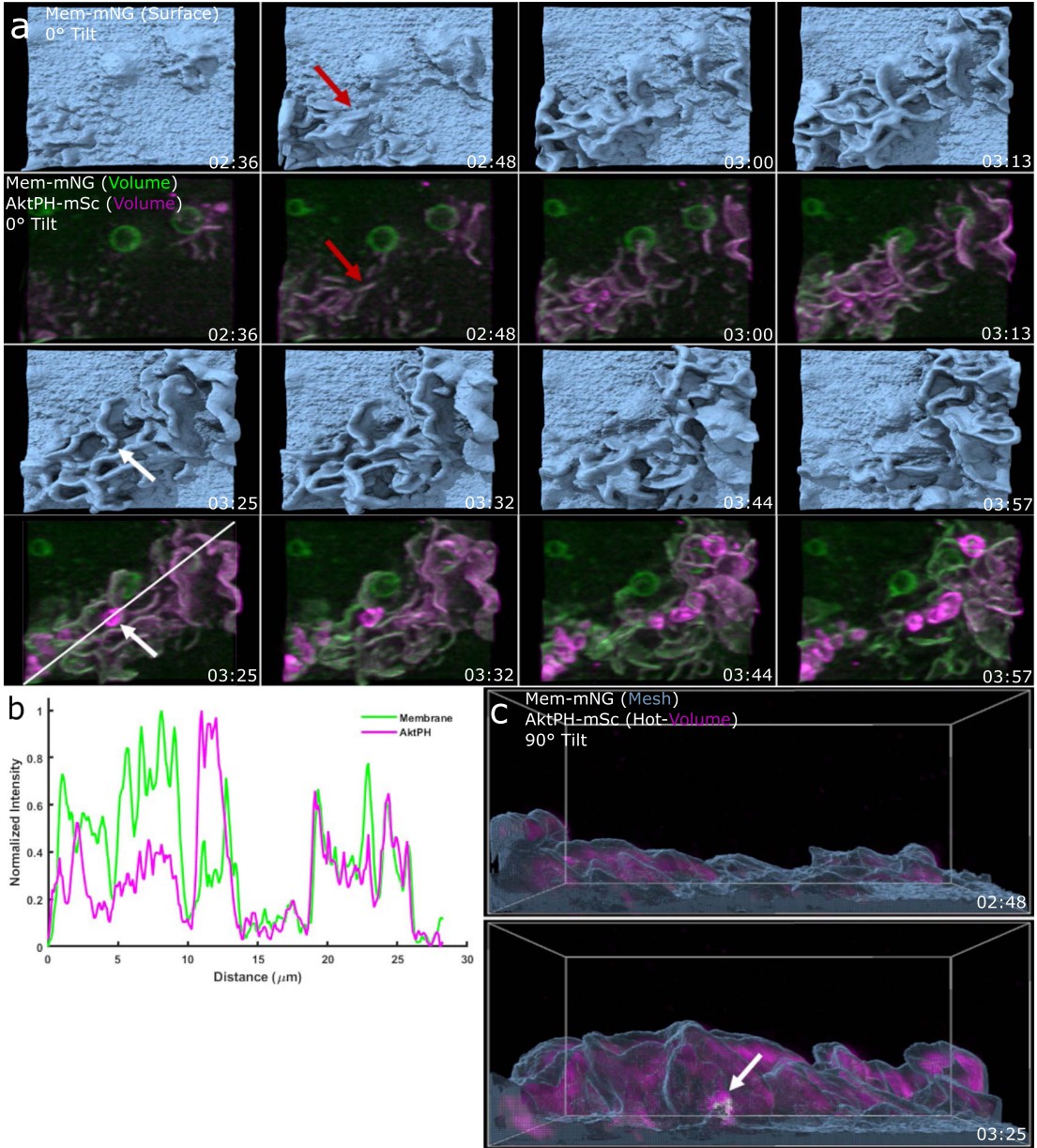

**Fig. 2 Early PI3K activity leads to amplification of PIP$_3$/PIP$_2$ in developing ruffles, macropinosome formation, and post closure recruitment. a** Top view of an Mem-mNG isosurface rendering provides depth for 3D visualization of ruffle extension. Dual-color volumetric intensity display comparing the recruitment of AktPH-mSc to early and expanding ruffles as well as sealed macropinosomes (region 21 × 19 μm x, y). **b** Intensity line scan of the volumetric Mem-mNG and AktPH-mSc shows their co-scaled relative intensities for extending membrane ruffles, as well as recruitment around a sealed macropinosome. **c** Side view of the isosurface mesh plasma membrane and volumetric AktPH-mSc (magenta hot color scale) from **a** shows that the early stages of ruffle development is filled with AktPH-mSc and the resulting macropinosome (white arrow) receives a final intense AktPH-mSc recruitment around the formed macropinosome at the bottom of the ruffle (region 21 × 19 × 15 μm).

from the plasma membrane (Fig. 2a) until maturation where tubulation and fusion between adjacent macropinosomes occurs (Supplementary Movie 3). As these early ruffles expanded laterally along the plasma membrane and protruded vertically from the cell surface, some ruffles continued to accumulate AktPH-mSc, whereas others lost AktPH-mSc and receded back into the cell suggesting that different levels of PI3K activity in neighboring ruffles influences the outcome of a ruffling region (Fig. 2b). Ruffles that maintained AktPH-mSc throughout the ruffle volume continued to grow and formed macropinosomes, which were accompanied by an intense transient recruitment of AktPH-mSc to the base of the ruffle around the nascent macropinosome

(Fig. 2c and Supplementary Movie 4). Given the early localization and amplification of PI3K signaling in ruffles that become macropinosomes, we wondered in what ways PI3K activity contributed to 3D ruffle dynamics.

**PI3K activity is required for macropinosome sealing, but not ruffling.** To gain insight into the role of PI3K in regulating the morphological dynamics of macropinocytosis, we used the broad-spectrum PI3K inhibitor LY294002, which inhibits the closure phase of macropinocytosis in macrophages[27]. Non-treated control cells formed transient dorsal ruffles that recruited AktPH-mSc and closed into macropinosomes, as seen by the surface rendering and

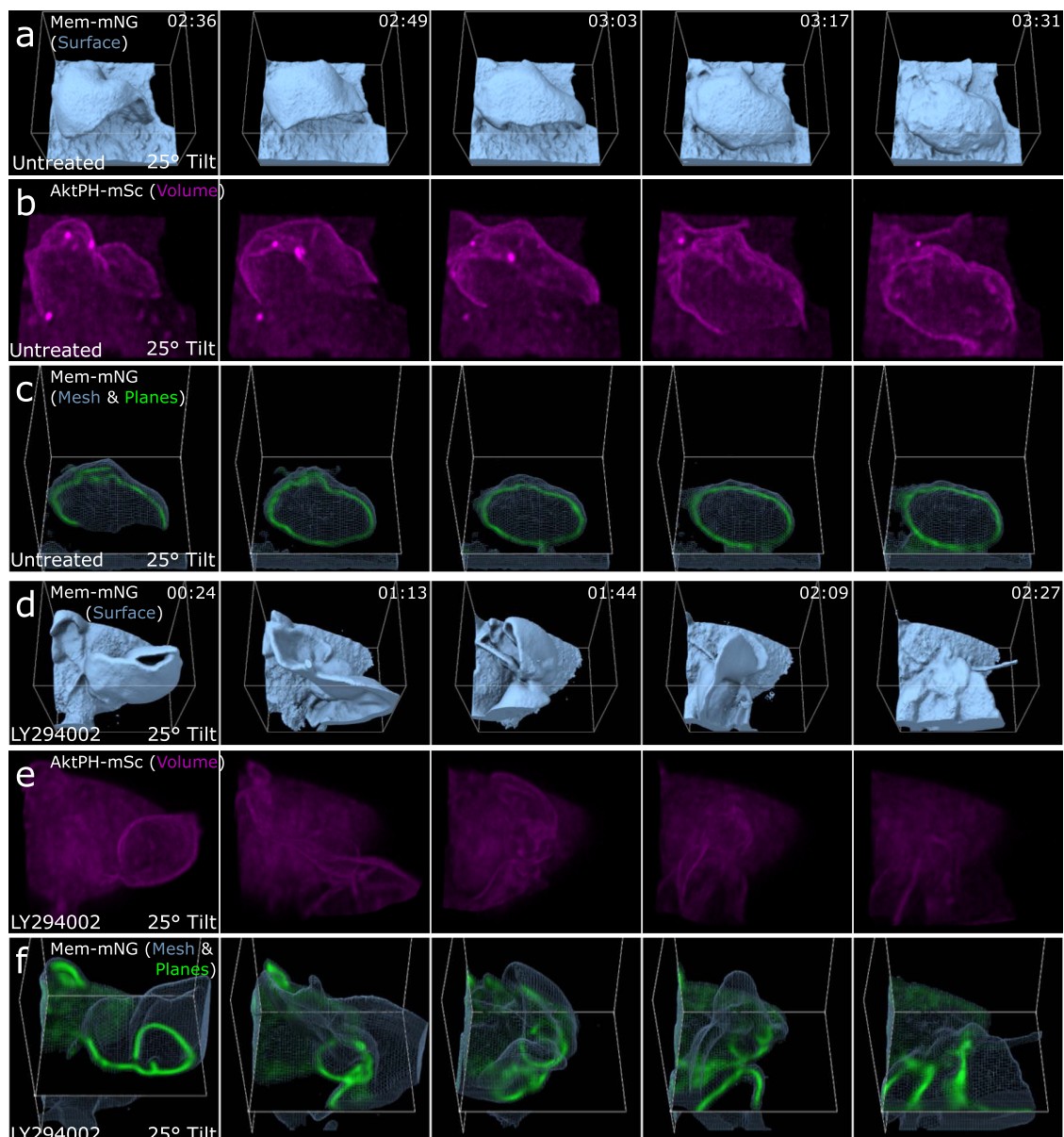

**Fig. 3 PI3K activity is required for membrane sealing and separation from PM/internalization of a complete macropinosome but not membrane ruffling. a** Isosurface rendering of Mem-mNG for an untreated macrophage during a successful macropinocytosis event where the sheet curls back toward the membrane for fusion/sealing (region 12 × 13 × 10 μm). **b** Volumetric rendering of Sc-AktPH of the untreated cell shows the increase of PI3K activity in the ruffle that creates a macropinosome (region 12 × 13 × 10 μm). **c** Mesh and orthogonal planes of Mem-mNG show the internal membrane organization of the ruffle and resulting macropinosome (region 12 × 13 × 10 μm). **d** Isosurface rendering of an LY294002-treated macrophage provides depth to the attempted closure of a macropinocytic cup (region 10 × 12 × 10 μm). **e** Volumetric intensity rendering of Sc-AktPH for an LY294002-treated macrophage shows the diffuse distribution of AktPH and minimal PI3K activity. The cytosolic intensities were co-scaled for the untreated and treated macrophage (region 10 × 12 × 10 μm). **f** xy-plane for the Mem-mNG probe of an LY294002-treated cell during a failed macropinocytosis event. In the surface view, the ruffle appeared to form a macropinosome; however, when overlaid with the plane view it became clear that it failed to fully form into a macropinosome. The ruffle quickly reduced in size and became undistinguishable within the cytosol, while never receiving the post closure increase of PI3K activity (region 10 × 12 × 10 μm).

intracellular void that is maintained in the plane view (Fig. 3a–c and Supplementary Movie 5). LY294002 treatment for 30 min did not stop ruffle formation but eliminated AktPH-mSc recruitment to membrane ruffles (Fig. 3d, e and Supplementary Movie 6). Furthermore, LY294002-treated cells formed ruffles that appeared to close into a macropinosome but retracted back to the plasma membrane and failed to maintain an intracellular organelle (Fig. 3f). In order to further understand the dynamic cellular response to

PI3K inhibition, we imaged the same cells immediately before and after LY294002 treatment. Immediately after treatment, PI3K activity was halted as evidenced by clearance of AktPH-mSc from the PM ruffles and diffuse cytosolic localization (Supplementary Fig. 2). Interestingly, many of the ruffles receded into the plasma membrane, but ruffling resumed within 10 min and was similar to the untreated cells by 30 min (Supplementary Movie 7). Taken together, these data suggest that PI3K activity is dispensable for

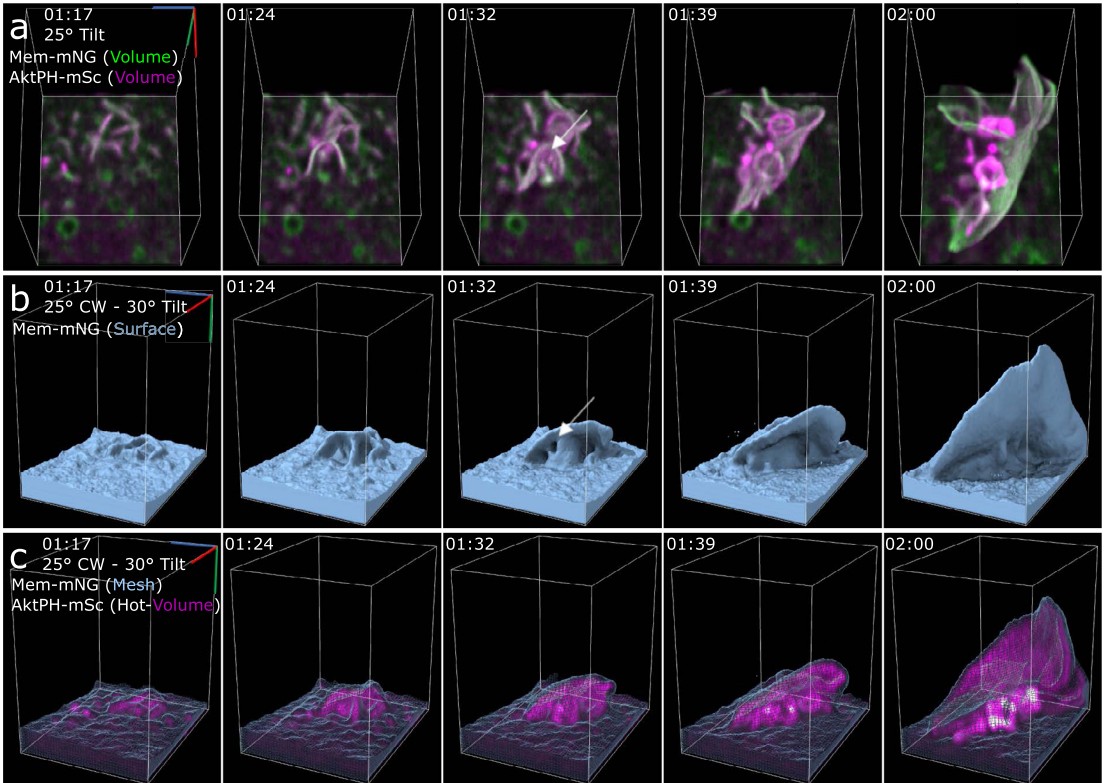

**Fig. 4 Macropinosomes form via PI3K-primed ruffle fusion. a** Dual volumetric intensities of Mem-mNG and AktPH-mSc show the intensity of each probe as the ruffles and macropinosomes form. The montage shows the earliest stage of the ruffle that extends vertically and forms macropinosomes along the length near the base of the primary ruffle as a result of smaller AktPH-mSc-rich extensions colliding. The white arrow points at the macropinosome-forming region further emphasized in the isosurface (region 9 × 12 × 13 μm). **b** Isosurface rendering of Mem-mNG shows the structure of the extending ruffle and the continued sheet extension after the macropinosomes formed. The white arrow emphasizes the small pocket that closes to form one of the macropinosomes (region 9 × 12 × 13 μm). **c** Mesh surface rendering of Mem-mNG and volumetric AktPH-mSc shows the internalized macropinosome with the increased localization of AktPH-mSc at the bottom of the ruffle (region 11 × 9 × 12 μm).

ruffle formation but is required for membrane sealing to generate a macropinosome.

**PI3K activity primes ruffles for fusion to seal into nascent macropinosomes**. We next sought to categorize the macropinosome formation based on the way the membrane fused and the relative amount of PI3K activity. Based on the previously established models for macropinosome formation, there would be distinguishing characteristics depending on the sealing formation method. Either we would find linear extensions where the distal tips would collide, circularize, and seal or we would find filopodial spikes that form and the membrane would fill in the space between before twisting to seal[12]. Surprisingly, we found that approximately 88% of the quantified macropinosomes formed when the leading edge of extending ruffles collided along the sides of nearby membrane surface or ruffles that typically only involved a small portion of the second ruffle, so long as the ruffle area had elevated AktPH-mSc (Fig. 4). The remaining 12% of events we observed were classified as tidal wave-like structures in which a mostly isolated planar ruffle extended from the cell surface where the entire ruffle was rich in AktPH-mSc, the ruffle gained curvature in a rolling fashion, and resulted in fusing back with the plasma membrane rather than an extending ruffle (Supplementary Movie 5). However, given that the entire ruffle area was rich in AktPH-mSc, these types of formation follow the same underlying mechanism as collisions with adjacent membrane extensions. Frequently, a single ruffle area produced multiple

macropinosomes and were the result of similar but smaller ruffle extensions that quickly fused near the base of larger ruffles (Fig. 4). Within the ruffle, forming macropinosomes recruited AktPH-mSc near the base of the ruffle as they transitioned into a spherical shape prior to detaching from the plasma membrane and moving independently (Fig. 4 and Supplementary Movies 3 and 8). We hypothesized that regions with highly concentrated AktPH-mSc localization would correlate with increased macropinocytic activity. Indeed, this phenomenon was observed in 4 of the 11 constitutive cells (Fig. 5). These ruffling regions resulted in the formation of many macropinosomes through the intersection of multiple ruffles that were nearly indistinguishable from one another and only became apparent through the PI3K post closure activity (Fig. 5d, e). Therefore, the elevated PI3K activity creates a microenvironment suited for the rapid fusion of PI3K-primed ruffles into macropinosomes of various sizes within short time-frames (Supplementary Movies 9 and 10). We hypothesized that other signaling that activates PI3K activity may stimulate distinct ruffling morphologies or modulate the rate of macropinocytosis.

**CSF-1 growth factor signaling promotes extensive circular ruffling and macropinocytosis**. CSF-1 is an essential macrophage growth factor that stimulates macropinocytosis at levels controlled by the concentration of the CSF-1 signal[9]. Macrophages starved of CSF-1 for 24 h and then acutely stimulated produced expansive circular ruffles that initiated from the distal cellular margins coincident with cellular spreading (Fig. 6). LLSM

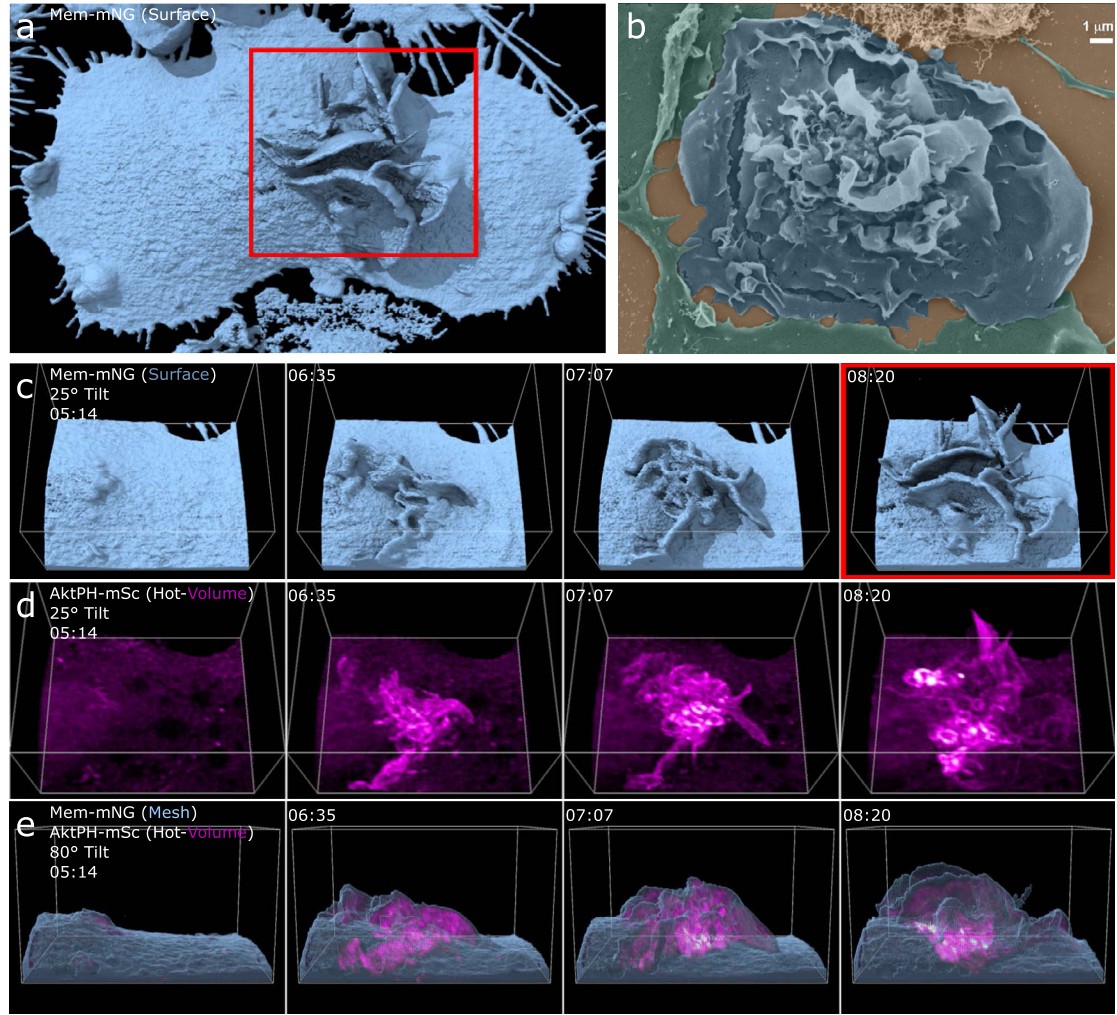

**Fig. 5 Phosphatidylinositol localization and chaotic ruffling underlie macropinocytosis in complex membrane structures. a** Single time point, full cell surface rendering of chaotic macropinocytosis event. The red box correlates to the same frame in **c**, **d**. **b** SEM images of a BMDM showing similar highly active ruffling regions. ($n = 10$ micrographs, scale bar 1 μm). **c** Isosurface montage shows the chaotic orientation of membrane structure (region 27 × 22 × 16 μm, 25° tilt). **d** Volumetric AktPH-mSc (magenta hot) provides a more detailed emphasis on the AktPH activity within the membrane ruffles and highlights the macropinosomes that have formed (region 27 × 22 × 16 μm, 25° tilted). **e** Mesh surface with AktPH-mSc (magenta hot) shows the AktPH activity as the ruffle develops as well as the increased recruitment around formed macropinosomes at the base of the event.

imaging revealed a circular ruffle that initiated at the edge of the cell with a height of approximately 2 μm above the dorsal surface and constricted to a central location in a coordinated manner (Fig. 6a). A striking feature of this ruffle was the confinement of AktPH-mSc within the limiting edge of the ruffle. As the circular ruffle constricted toward the center of the cell, AktPH-mSc was highly concentrated within it and was nearly undetectable in the rest of the cell (Fig. 6b and Supplementary Movie 11). In the volumetric projections, many macropinosomes were observed to form during the constriction process without additional membrane protrusions being generated. This is consistent with our hypothesis that increased PI3K activity primes membranes for fusion to generate macropinosomes (Fig. 6c a and Supplementary Movies 11 and 12). Thus, CSF-1 initiated long-range signaling and PI3K activation resulting in coordinated movements of the cytoskeleton throughout the cell.

**LPS stimulates regional ruffling and generates large numbers of macropinosomes.** The bacterial cell wall component LPS activates PI3K through the Akt pathway[28] and acutely stimulates macropinocytosis[3]. Recently, LPS stimulation was used to

characterize a novel formation mechanism involving actin tent-poles supporting membrane veils that cross to create a macropinosome[12]. When FLMs were exposed to LPS, regional patches of membrane ruffling were generated that migrated around the dorsal surface of the macrophage (Fig. 7a and Supplementary Movie 13) in a manner distinct from the dorsal surface ruffle generated by CSF-1-stimulated cells (Fig. 6); however, this process was similar in appearance to constitutive macropinocytosis (Fig. 7c and Supplementary Movie 14). The patches of ruffles in LPS-treated cells generated many small ruffles, had elevated PI3K activity, and were more efficient at forming macropinosomes as compared to control (Fig. 7d). Thus, the nature of macropinosome formation is coordinated over different length scales with differing intensities depending on the nature of the activating stimulus. Regardless, PI3K activity delineates ruffles and regions of the plasma membrane where macropinosomes form.

**Discussion**
Here we have utilized LLSM to develop a new level of understanding of the structural dynamics and PI3K signaling

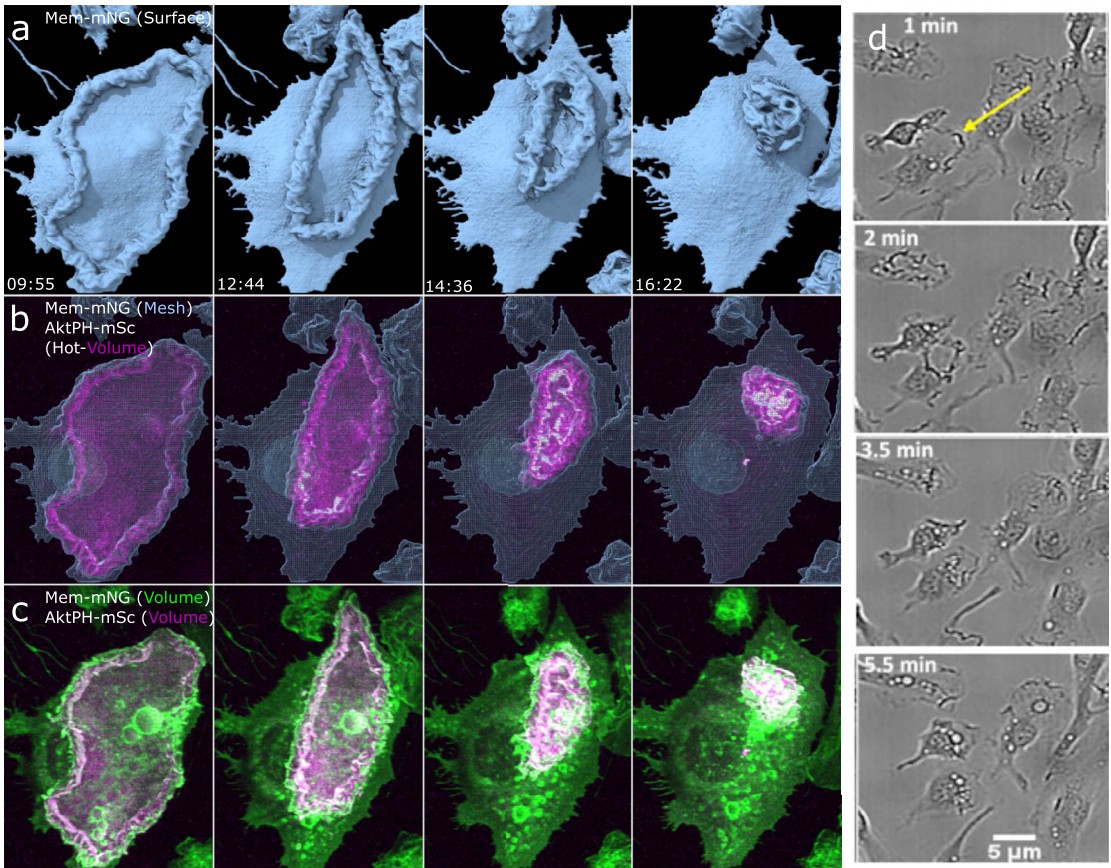

**Fig. 6 Growth factor starvation and stimulation results in the formation of large circular dorsal ruffles that corrals PIP₃/PIP₂.** Macrophages were starved of CSF-1 for 24 h, imaged for 5 min as a baseline, and imaging restarted 1 min after stimulation with CSF-1. Four-frame montages provide a visual display of the large dorsal ruffle that acts as a diffusional barrier that restricts PIP₃/PIP₂ to the inside of the ruffle as it is cleared from the surface. This barrier is likely acting as a signal amplification mechanism stimulating the production of many macropinosomes. **a** Isosurface rendering provides crisp surface directionality; **b** surface mesh and volumetric AktPH-mSc (magenta hot) show the restricted probe as the membrane converges. **c** Volumetric intensity of both Mem-mNG and AktPH-mSc show the intensity locations of the membrane ruffle and the restricted AktPH (49 × 60 μm x, y). **d** Bright field images showing multiple cells responding to stimulation with similar dorsal membrane clearing.

underpinning macropinocytosis. Until recently, dynamic processes such as macropinocytosis were characterized using optical techniques with poor axial resolution and elevated phototoxicity leading to subsampling the spatial and temporal dynamics and requiring inference from multiple methods such as scanning electron microscopy of fixed cells to address the formation mechanism of macropinosomes. Light sheet microscopy overcomes these obstacles and enables us to record, with sufficient spatial and temporal resolution, the complete evolution of membrane ruffles and the mechanism by which these ruffles form into macropinosomes while also measuring the redistribution of signaling molecules controlling these processes. We have shown that macropinosomes form through several possible morphologies; however, in each case PI3K activity primes ruffles for fusion with adjacent primed membranes to form macropinosomes. In areas with elevated PI3K activity, either naturally or through external stimulation, there was an increased ruffle density leading to an increased probability of primed ruffles colliding to form macropinosomes. This model of macropinosome formation relying on PI3K priming rather than a defined geometry also explains the variation in diameter that is a hallmark of macropinosomes. The improved sensitivity of LLSM enabled detection of PI3K activity at the earliest stages of ruffle development that grows in curving ruffles and peaks around macropinosomes post closure. Our data are consistent with a

mechanism driven by the geometry of curving ruffles that confines PI3K, thereby amplifying the signal, which in turn activates yet unknown fusogenic protein(s) localized to the ruffle edges mediating sealing during membrane collisions. This conclusion is supported by the observations that inhibition of PI3K activity with LY294002 did not substantially alter membrane ruffling structure, curvature, or collisions but completely inhibited sealing, even when fully spherical morphologies were observed that then collapsed back into the cell surface. These data are consistent with previous work using LY294002 as a broad-spectrum PI3K inhibitor and the extensive work done on PI3K in *Dictyostelium*[29,30].

The model suggested in this work contrasts with a recent report in LPS-activated RAW 264.7 cells that described F-actin-rich filopodial "tentpoles" protruding from the surface that twisted to constrict veils of membrane that then became macropinosomes[12]. Using Mem-mNG and Lifeact-mSc in RAW 264.7 cells under identical imaging conditions, we observed filapodial extensions that fit with the tentpole formations described previously (Supplementary Fig. 3 and Supplementary Movies 15). However, we also transduced BMDM with the same Mem-mNG probe and found no tentpole-like structures; rather the FLM and BMDM were similar in their flat and smooth dorsal surfaces with large lamellar sheet-like ruffles (Supplementary Fig. 3 and Supplementary Movies 16), suggesting that the tentpole-like

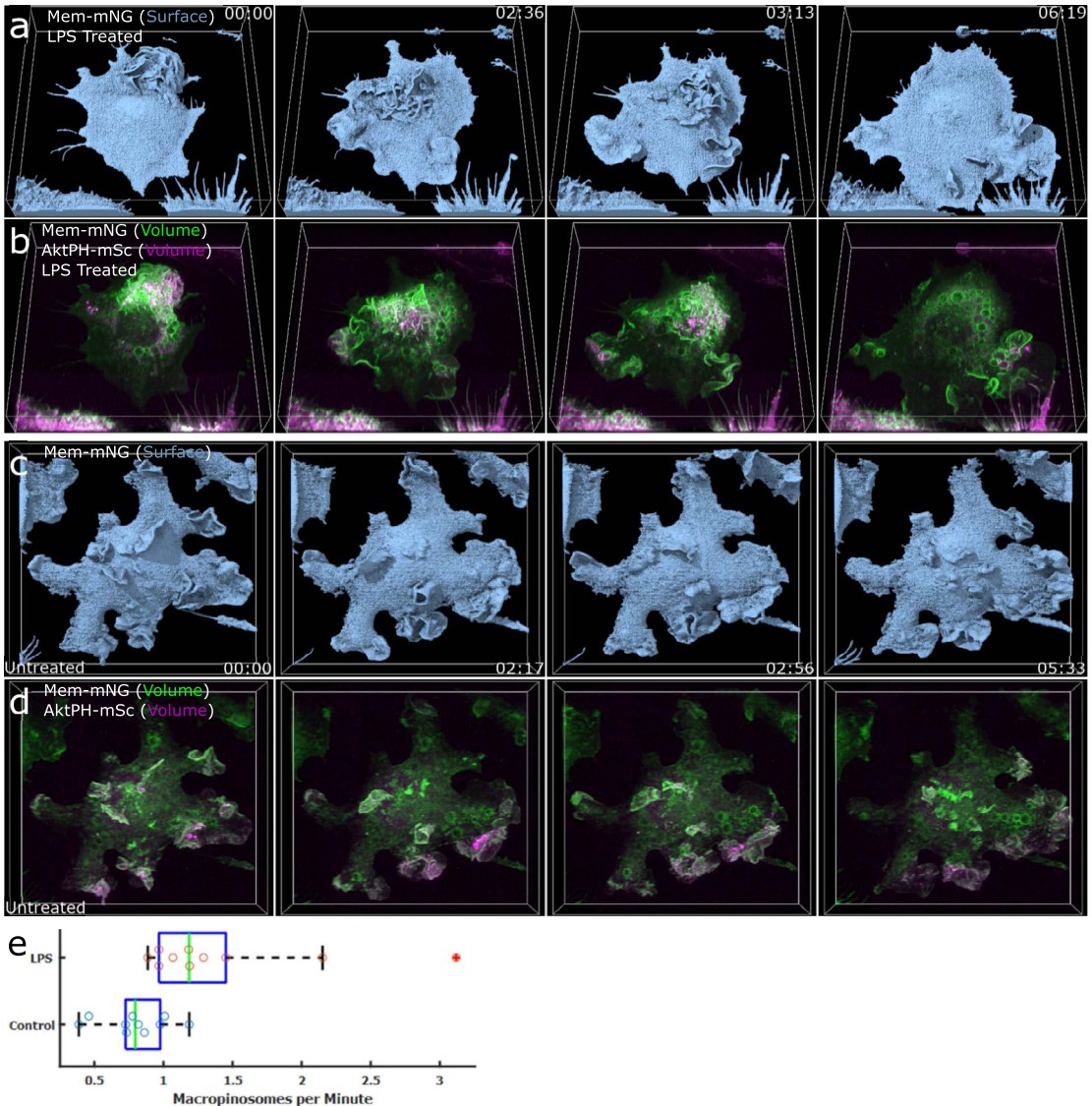

**Fig. 7 LPS stimulation increases membrane ruffling and macropinocytosis.** Macrophages were treated with 100 ng ml$^{-1}$ LPS for 24 h prior to imaging. **a** Surface rendering of Mem-mNG on an LPS-stimulated macrophage provides a surface-level understanding of the membrane, exploration, ruffling, and PM structure (region 68 × 69 × 16 μm). **b** Dual-color volumetric intensity projections of Mem-mNG and AktPH-mSc for an LPS-stimulated cell provided the intensity activity during increased macrophage activity and shows the highly AktPH-rich regions of membrane ruffling (region 68 × 69 × 16 μm). **c** Untreated macrophage isosurface displaying less exploratory behavior (region 68 × 64 × 16 μm). **d** Dual-color volumetric intensity rendering of the untreated macrophage gives insight on the AktPH activity inside of the cell during macropinocytosis and allows for the quantitative comparison of macropinosomes formed between the stimulated and unstimulated cells (Region 68 × 64 × 16 μm). **e**) Differences in macropinocytic activity between untreated and LPS-treated macrophages. Box plot showing median (green line), IQR (blue box), range (dashed line) without outlier (asterisk), $n = 10$ biologically independent control cells examined over 7 independent coverslip experiments (blue circles), and $n = 10$ biologically independent LPS-stimulated cells examined over 5 independent coverslip experiments (red circles). All macropinosomes >1 μm were manually counted using a z-projection MIP in Fiji and were distinguished by the post closure spike in AktPH-mSc intensity.

formation is dispensable for macropinocytosis and may be unique to the RAW 264.7 cell line.

Our data demonstrate that macropinosomes form from a variety of morphologies in macrophages, contrary to the morphologies in other cells such as the tightly controlled process observed in *Dictyostelium*. Specifically, macropinocytosis in *Dictyostelium* appears to involve the formation of a cup with a clearly segregated PIP3-containing domain[29], whereas our observations as well as the work of Condon et al. using LLSM indicates that initiation of macropinosomes by macrophages arises from a morphologically diverse and somewhat chaotic ruffling process. Importantly, the well-defined PIP3-containing domains observed

in *Dictyostelium* appear similar to the macropinosomes in macrophages at the sealing stage, in which substantial PIP3 products accumulate (Fig. 2). We speculate that these differences are attributed to the importance of macropinocytosis in *Dictyostelium* as a method of nutrient gathering[30,31], while macrophages perform constitutive macropinocytosis in order to sample the environment for antigen presentation[2,7].

Taken together, our experiments indicate a mechanism for macropinosome formation requiring amplified PI3K signaling within ruffles that become macropinosomes and that PI3K contributes primarily to priming membranes for sealing. The membrane probe and visualizations we have described set the

foundation to enable rigorous testing of this mechanism using specific inhibitors and sensors that bind to the various products to determine how each step is regulated during macropinocytosis.

## Methods

**Plasmids.** pCMV-VSV-G was a gift from Bob Weinberg (Addgene plasmid #8454; http://n2t.net/addgene:8454; RRID:Addgene_8454)[32]. psPAX2 was a gift from Didier Trono (Addgene plasmid #12260; http://n2t.net/ addgene:12260; RRID: Addgene_12260). pLJM1-EGFP was a gift from David Sabatini (Addgene plasmid #19319; http://n2t.net/addgene:19319; RRID:Addgene_19319)[33]. Lck-mScarlet-I and Lifeact-mScarlet-I were gifts from Dorus Gadella (Addgene plasmids #98821, #85056; http://n2t.net/addgene:98821; http://n2t.net/addgene:85056; RRID: Addgene_98821, RRID:Addgene_85056)[34].

**Construction of the membrane and AktPH probes.** The membrane probe was constructed by combining the membrane localization motif (MGCVCSSNPE) from Lck[34] in frame with mNeonGreen in the pLJM1 backbone containing the puromycin resistance gene. The AktPH-mSc probe was constructed by using the pleckstrin homology domain from Akt in frame with mScarlet-I in the pLJM1 backbone, modified to contain the blasticidin resistance gene. Sequences were codon optimized for expression in mouse cells, synthesized, and sequence verified by GenScript (Piscataway, NJ). The plasmids have been deposited to Addgene.

**Viral transduction.** Sequence-verified plasmids containing genes encoding FP-chimeras plus the packaging plasmids pVSV-G and psPAX2 were transfected into NIH 293T cells for packaging using linear 25 kDa polyethylenimine as a transfection reagent. NIH 293T cell culture supernatant containing lentiviral particles was collected and added to BMDMs, RAW 264.7, or FLMs each treated with cyclosporin A (10 μM) for 2 days. Transduced FLMs were selected with puromycin and blasticidin (10 μg ml⁻¹ each).

**Macrophage culture media.** Dulbecco's Modified Eagle Medium (DMEM)/F-12 (Gibco) supplemented with 20% heat-inactivated fetal bovine serum (FBS; R&D Systems), 1% penicillin/streptomycin (Gibco), 50 ng ml⁻¹ mCSF-1 (BioLegend), and 5 μg ml⁻¹ plasmocin (Invivogen) maintained at 37 °C with 7.5% CO₂.

**PIP₃ inhibition (LY294002).** Coverslips were prepared and moved to a well of media containing 50 μM LY294002 where it was incubated for 30 min at 37 °C and 7.5% CO₂ prior to imaging. The coverslip was then transferred to the LLSM bath containing Imaging Media, 1.7 mM glucose, and 50 μM LY294002. The coverslip was explored using the LLS software, and three cells were chosen per coverslip that provided the best visual representation of the population. Each cell was imaged under the same parameters as described in the "Lattice light sheet microscopy" section. Additionally, coverslips were tested to ensure that the 30 min treatment was sufficient by imaging a coverslip, introducing LY294002 at 50 μM, and imaged three cells representing the full 30 min treatment. Finally, untreated coverslips were placed on the microscope and three cells were selected. The three cells were imaged prior to drug treatment in the imaging bath, then LY294002 was introduced at 50 μM and each cell was reimaged in reverse order (1-2-3-3-2-1). Alongside these tests, a DMSO control was performed to further ensure the proper treatment.

**CSF-1 stimulation.** The coverslip was starved overnight in DMEM/F-12 with 10% FBS and 1% pen/strep. After 24 h, the coverslip was moved to the LLS bath containing 7 ml of L-15 imaging media, and 1.7 mM glucose. The coverslip was explored using the LLS software, and three cell targets were chosen and imaged for a pre-stimulation comparison. Immediately after imaging the third cell, CSF-1 was introduced at 50 ng ml⁻¹ to the 7 ml bath. The third cell was once again imaged <1 min after stimulation and each additional cell was imaged in reverse order (Imaging order 1-2-3-3-2-1).

**LPS stimulation.** FLMs were stimulated with 100 ng ml⁻¹ LPS from *Salmonella enterica* serotype enteritidis (Sigma) for 24 h in culture media before being transferred to imaging media containing 100 ng ml⁻¹ for LLSM experiments.

**Macrophage isolation.** FLM cell cultures were generated as described previously[35,36]. Livers were isolated from gestational day 15–19 mouse fetuses from C57BL/6J mice (The Jackson Laboratory, Bar Harbor, ME) in accordance with South Dakota State University Institutional Animal Use and Care Committee. Liver tissue was mechanically dissociated using sterile fine-pointed forceps and a single-cell suspension was created by passing the tissue through a 1 ml pipette tip[35]. Cells were plated on non-tissue culture-treated dishes and kept in growth and differentiation medium containing the following: 20% heat-inactivated fetal bovine serum; 30% L-cell supernatant, a source of M-CSF[37,38]; and 50% Dulbecco's modified growth medium containing 4.5 g l⁻¹ glucose, 110 mg l⁻¹ sodium pyruvate, 584 mg l⁻¹ L-glutamine, 1 IU ml⁻¹ penicillin, and 100 μg ml⁻¹ streptomycin. FLM were cultured for at least 8 weeks prior to transduction and experiments.

**Cell culturing and coverslip plating.** FLMs were cultured in T-25 tissue culture flasks using the following culture media: DMEM F-12 with 20% FBS, 1% penicillin/streptomycin, 50 ng ml⁻¹ CSF-1, and 5 ng ml⁻¹ plasmocin. The cell cultures were split at ~85% confluence, first by washing the T-25 flask with 3 ml of phosphate-buffered saline (PBS) (−Ca/−Mg) 2 times. The cells were then lifted from the T-25 flask using 4 ml of 4 °C PBS (−Ca/−Mg, +0.98 mM EDTA) with gentle pipet washing for approximately 10 min. The lifted cells were moved to a 15 ml centrifuge tube (1 ml of culturing media was added to 15 ml tube if cells took >10 min to lift) and centrifuged at 200 × g for 5 min. While the cells were being spun down, the T-25 flask was washed with PBS (−Ca/−Mg), filled with 5 ml of culture media, and placed in the 37 °C incubator to reach appropriate culture conditions. Once the cells were finished being spun down, the supernatant was removed from the 15 ml tube and the cells were resuspended in 1 ml culturing media and counted. The counting was done by mixing 10 μl of suspended cells with 10 μl of trypan blue and placed on a glass slip to be counted using a countess. The FLMs were re-plated in the original T-25 flask with ~6–7 × 10⁵ cells. The cells were washed every 2 days and given fresh media until reaching ~85% confluence where they would then be split. Cell lines were kept for approximately 2 months before being replaced with early state frozen aliquots.

Macrophages were prepared for LLS imaging 24 h prior to imaging using 5 mm glass coverslips. The coverslips were soaked in 90–100% ethanol and flame cleaned using a butane flame. Approximately 5 flame cleaned coverslips were placed per well of a 12-well plate each containing 1 ml of culture media. Cells were added to each well during the cell culture process at ~3 × 10⁵ cells in each 3.5 cm² (12-well plate) for imaging. The FLMs were incubated on the flame-cleaned glass coverslips in culturing media for 24 h prior to imaging. The coverslips were transferred to the LLSM bath that was filled with 7 ml of Leibovitz's L-15 Media (supplemented with 1.7 mM glucose) at ~37 °C.

The RAW 264.7 macrophages were purchased from ATCC and maintained using the suggested subculture routine. Briefly, cells were split between 70 and 80% confluence, reseeded at 2–4 × 10⁴ cells cm⁻², and cultured in DMEM F-12, requiring 7.5% CO₂ at 37 °C. Due to the variability in adherence, cells were lifted using the recommended cell scraper protocol. The same protocol as discussed above was used for plating cells for imaging on the LLSM.

**Lattice light sheet microscopy.** The LLSM is a replica of the design described by Chen et al.[25], built under license from HHMI. Volumetric image stacks were generated using dithered square virtual lattices (Outer NA 0.55, Inner NA 0.50, approximately 30 μm long) and stage scanning with 0.5 μm step sizes, resulting in 254 nm deskewed z-steps. Excitation laser powers used were 18 μW (488 nm) and 22 μW (561 nm), measured at the back aperture of the excitation objective. The emission filter cube (DFM1, Thorlabs) comprised a quadband notch filter NF03-405/488/561/635 (Semrock), longpass dichroic mirror Di02-R561 (Semrock), shortpass filter 550SP (Omega) on the reflected path, and longpass filter BLP01-561R (Semrock) on the transmitted path; the resulting fluorescence was imaged onto a pair of ORCA-Flash4.0 v2 sCMOS cameras (Hamamatsu). The camera on the reflected image path was mounted on a manual x–y–z translation stage (Newport 462-XYZ stage, Thorlabs), and the images were registered using 0.1 μm Fluoresbrite YG microspheres (Polysciences). The image capturing rates varied between 5 and 10 s per volume using 8–12 ms planar exposures depending on the brightness of the cell and imaging region.

**Scanning electron microscopy.** BMDMs were plated onto 13 mm diameter circular glass coverslips and cultured overnight in RPMI with 10% FBS (R10). To stimulate macropinocytosis, BMDM were incubated 30 min in PBS, then 15 min in PBS containing 10 nM CSF-1. Cells were fixed in 2% glutaraldehyde, 0.1 M cacodylate buffer, pH 7.4, 6.8% sucrose, 60 min at 37 °C. Fixative was replaced with a second fixative consisting of 1% OsO₄ in 0.1 M cacodylate buffer, pH 7.4, for 60 min at 23 °C. The second fixative was replaced with 1% tannic acid in cacodylate buffer, 30 min at 4 °C, then rinsed with 3 changes of 0.1 M cacodylate buffer. Coverslips were transferred through successive changes of acetone–water mixtures, progressively increasing acetone concentrations to 100% before a final incubation in hydroxymethylsilazidane (HMDS; EM Sciences). HMDS was removed and coverslips were dried for 2 days. Coverslips were shadowed with gold and observed on an Amray 1900 field-emission scanning electron microscope.

**Deconvolution and post processing.** The raw volumes acquired from the LLSM followed the standard protocol, including deskewing, deconvolving, and rotating to coverslip coordinates in LLSpy[39] as the raw data acquired from LLSM is captured at an approximate angle of ~32° with respect to the coverslip. We applied a fixed background subtraction based on an average dark current image, 10 iterations of Lucy–Richardson deconvolution with experimentally measured point spread functions for each excitation followed by rotation to coverslip coordinates and cropping to the region of interest surrounding the volume for visualization. We optimized the illumination for minimal photobleaching (<5% of initial) during the experiments. The fully processed data were opened as a volume map series in UCSF ChimeraX and utilized isosurface, mesh, 3D volumetric intensities, and orthogonal planes renderings to examine the data. The surface and mesh options utilize a 3D analog of an isoline called an isosurface that represents points in

volume space as constant values that were used to display the membrane probe. The isosurface provides a defined surface for the membrane probe resulting in shadowing providing visual depth to the 3D data. The mesh rendering offers a similar surface definition while also providing an option to include the internal fluorescent AktPH-mSc signal. We used a volumetric intensity projection to visually display the localization of AktPH-mSc throughout the cell. The final technique used to display the LLSM data was through orthogonal planes. This generates 2D planes each 0.103 μm thick for the entire volume in *xy*, *yz*, and *xz*. Multiple methods were overlapped and shown side by side to effectively represent the data and labeled within each figure. All relevant custom codes used to generate the figures and videos have been included as a Supplementary File.

**Reporting summary**. Further information on research design is available in the Nature Research Reporting Summary linked to this article.

## Data availability

The processed LLSM data along with one version of raw data are available through the Cell Image Library (CIL) database under accession codes CIL: 54592–54604 available at http://cellimagelibrary.org/groups/54592. Source data are provided with this paper.

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

## Acknowledgements

Funding was provided by the South Dakota Board of Regents through BioSNTR and the SDBOR FY20 collaborative research award "IMAGEN: Biomaterials in South Dakota" (to S.S.). Additional funding was provided by the National Science Foundation through research award CNS-1626579 "MRI: Development of a Scalable High-Performance Computing System in Support of the Lattice Light-sheet Microscope for Real-time Three-dimensional Imaging of Living Cells" (to R.B.A.). J.A.S. was supported by NIH grant R35 GM131720. B.L.S. is supported by the Chan Zuckerberg Initiative through the Imaging Scientist program (to S.S.). The data visualization and analyses were performed using UCSF ChimeraX, developed by the Resource for Biocomputing, Visualization, and Informatics at the University of California, San Francisco, with support from National Institutes of Health R01-GM129325 and the Office of Cyber Infrastructure and Computational Biology, National Institute of Allergy and Infectious Diseases. The Lattice Light Sheet Microscope referenced in this research was developed under license from Howard Hughes Medical Institute, Janelia Farms Research Campus ("Bessel Beam" patent applications 13/160,492 and 13/844,405).

## Author contributions

S.E.Q. acquired and analyzed LLSM data and co-wrote the manuscript. L.H. generated FLM cell lines used in experiments. J.G.K. designed plasmid constructs and generated FLM cell lines. J.A.S. performed SEM experiments and edited the manuscript. S.S. provided supervision and edited the manuscript. A.D.H. co-wrote the manuscript. R.B.A. built the LLSM and analysis cluster and provided supervision. N.W.T. designed experiments and co-wrote the manuscript. B.L.S. designed and performed initial experiments, assisted with data analysis, and co-wrote the manuscript.

## Competing interests

The authors declare no competing interests.
