## [Peer Review File · Nature Communications]

REVIEWER COMMENTS

Reviewer #1 (Remarks to the Author):

The uptake of fluid into macropinocytic vesicles is readily visible under the light microscope, but due to the scale and rapid movement of macropinocytic structures, it is difficult to describe in any detail by wide-field, confocal or spinning disc microscopy. Thus the morphologies that produce macropinosomes are not yet well established. Three have been suggested: cells project cups from their plasma membrane that contract at the lip and seal to release a macropinocytic vesicle into the cytoplasm (ref 13); or they form flaps that collapse back to the surface and fuse with it, trapping a droplet of medium; or finally they support and close cups using tentpoles (ref 12).

In contrast, lattice light sheet microscopy is perfect for the job. This paper presents lattice light sheet movies and images of stunning beauty using dual reporters for the plasma membrane and PIP3/PI3,4P2, to follow the formation of macropinosomes in macrophages. It shows that they often form from ruffles that collapse back onto the cell surface, or collide with each other, and then seal together. This mode occurs in un-stimulated macrophages and after stimulation with either CSF1 or LPS. These observations suggest that the spatial organization of macropinosome is less demanding than previously thought, and that the whole ruffle surface may be fusogenic.

PI3-kinase plays a central role in organizing macropinocytosis in both macrophages and *Dictyostelium amoebae*, but exactly how remains unknown. This work shows that macropinocytic ruffles are enriched in PIP3, giving general support to its importance, and that inhibition of PI3-kinase with LY294002 still allows ruffles to form, but prevents their sealing into macropinosomes, confirming previous work, but at better resolution.

In summary, this is an exciting paper, which helps us understand how macropinosomes form and the role of PIP3. The microscopy is beautiful. Publication would certainly advance the field. However, there are some issues that need addressing, as below.

1. It would be helpful if the authors could enquire further into why there are differing descriptions of how macropinosomes form (cups, tentpoles, ruffles colliding). Perhaps all three routes are true in different circumstances or for different cells? Ideally, they should examine macrophages from the Swanson lab and RAW cells from the Stow lab in their conditions to test whether the differences are due to the different cell lines or conditions used previously.

2. Earlier work describes a stage before cup formation in which a linear ruffle circularizes and PIP3 accumulates within it (eg ref 22). Very similar images are produced here (eg movie 1, starting at 5:39), but now with the superior resolution of LLSM, are interpreted as being due to a ruffle folding back onto the cell surface and progressively fusing with it, followed by PIP3 accumulation on the sealed vesicle. Do the authors feel this is a reasonable interpretation, which might explain the earlier observations?

And separately, do the authors see macropinosomes forming from cups? And if so what is their frequency?

3. The concentration of LY294002 used – 0.16 μM – is surprisingly low. The IC50 for PI3-kinase activity is 1.4 μM and for fluid phase uptake by macrophages, 3 μM (ref 24). How is it known that PI3-kinase is significantly inhibited, let alone so strongly that it can be stated that ruffling is independent of PIP3? What do higher doses do? A caveat to these experiments is that LY294002 inhibits both TORC1 and TORC2 (eg EMBO J 15, 5256, 1996). It should also be noted that the importance of PIP3 is strongly supported by *Dictyostelium* genetics, which shows that PI3-kinases are essential for macropinocytosis, with one of them having a specific role in closure (J Cell Sci 126, 4296 2013).

3. An idea advanced from Dictyostelium is that PIP3 domains activate actin polymerization at their edges by recruiting the WAVE complex there (ref 19). The formation of a circular ruffle around the PIP3 domain in giant circular ruffles shown in this paper supports the idea. Is this a new observation, or has it been reported before (I feel it has)?

The Dictyostelium model predicts that the extending edge of the ruffle, where actin polymerization occurs, should be a PIP3 domain boundary and therefore that one membrane of the ruffle should have PIP3 and the other not. This looks not to be true in many images, but is supported by a few. Could this be investigated in more detail, if the resolution and sectioning allow?

4. Minor things: please show scale bars and explain isosurface and volumetric intensity projections in a few simple words. How is the moment when a macropinosome seals identified? Please mark some sealing events in the movies. The cells look very flat - roughly how thick are they? And do they look like this in vivo?

Reviewer #2 (Remarks to the Author):

The work presented by Dr. Scott and colleagues entitled "The structural dynamics of macropinosome formation and PI3-kinase-mediated sealing revealed by lattice lightsheet microscopy", which shows the biological findings presumably based on the live imaging analysis from LLSM featuring with higher axial resolution. Taking the advantages of the high spatiotemporal resolution and low sample damage from LLSM, they revisit the micropinocytosis in macrophage-like cell lines and shows that PI3K activity is amplified in ruffles and critical for sealing them into macropinosomes, less effect on the ruffling rate and morphology. They did a lot of the live imaging (122 macropinosome formations) and got the conclusions based on the visualization techniques from the 4D (xyz +t) data with and without drug treatment. A lot of efforts have been put for collecting the live imaging data in order to conclude their new findings, which is a hard work due to the complicated management for live imaging. The reviewer appreciates their job for multicolor 4D imaging analysis. Through the manuscript, they used a lot of descriptions to explain the image data along with time frame. Given that more quantitative analysis with time evolving, that will merit these 5D imaging data (x, y, z, t, λ). Overall, the manuscript is well-written and the figures support their findings, which is good for the publication. However, the reviewer would like to ask the authors to address the following concerns,

1. In the introduction part, line50-53, the authors make the arguments about the PI3K activity is important for macropinosome formation. But in the experimental results, they monitored AktPH, probing the products of PI3K. Although the AktPHJ seems has been used to characterize PI3K activity during micropinocytosis, the reviewer wonders whether direct watching the PI3K activity is feasible or not, that is probing PI3K. Moreover, the reviewer didn't see any immediately increase or decrease the PI3K activity or PI3P/PI2P methods in your manuscript. For example, the Rapamycin inducible dimerization system to increase or decrease the PI3P/PI2P.

2. For all the figures and imaging analysis, the processed or imaging rendering have been applied to the presented data, the reviewer still thinks that the raw data or live images should be given at least for one data set or presented figures. The rendering software ChimeraX, there are many parameters involved for the imaging rendering. Therefore, raw imaging should be given for clarification. Moreover, there is an intensity plot along the image such as in figure 2b, the review think this should be done in raw image instead of processed one.

3. In line114, the corresponding area of the macropinosome process also need to be marked in isosurface view in Figure 2a. In Figure 2b, the intensity plot is made from the 3D maximum intensity projection? Since this a 3D data, not sure this is a good way to present this? Since this is a time-evolving process, an intensity plot with "time" variable for the description for different levels of PI3K activity would probably be a better way.

4. In Fig. 3, the PI3K inhibitor LY294002 shows the different levels of effect on the macropinosomes formation. Presumably PI3K activity affects the sealing process. Although the isosurface view gives the good insight for the macropinosome formation, the presentation lacks the intensity information, especially for PI3K. If possible, the reviewer would like to see how the PI3K intensity changes during the macropinosome formation with time axis within the whole macropinosome by volume displaying with intensity colormap. That will make all these data more valuable. A related supplementary movie shows the intensity change of macropinosome will be better.
5. For drug treatment part, how long does PI3K inhibitor LY294002 take to affect the cells? According to the experimental parts in line 246, 30 mins pre-incubation is used. Could you do the live-cell imaging which including before and after drug dosing? That would solidify the idea about "PI3K activity primes ruffles for fusion to seal nascent macropinosomes". And if the removal of the drugs, do the cells recover from the effect? What's difference between CSF-1 and LPS in your system? Can you use same analysis tool (figure 7e) to analyze the CSF-1 treated cells?
6. For temporal resolution, while the review went through the movies, it seems that the membrane ruffling and macropinosomes formation have different rate. The initial step is slower and macropinosome is faster. It seems the equal time interval is used for the live imaging resulting in lacking details for the macropinosome formation. For example, supplementary figure 1 shows the constitutive micropinocytosis. The complicated structures formed at the lateral time points. If the better temporal resolution is applied, the fine details will be obtained for the macropinosome formation.
7. For spatial resolution, in figure 5, there is a nice image for SEM, but no scale bar information. The reviewer would like to see the thickness of ruffling membrane could be measured or not? That probably gives you the sense about your optical microscope image vs high resolution SEM image. Especially, your 3D surface rendering way will broad your spatial resolution.
8. The other concern is about the membrane density vs PI3K density. In the figures, the PI3K shows the high intensity at the base of macropinosome, is it possible due to the accumulation of membrane.
9. Since the work present the time evolving data for micropinocytosis event, again the reviewer hopes to see the 3D time profiles of intensity and morphology for the event and by doing this will merit the high spatiotemporal resolution of LLSM.

In response to the helpful suggestions by the reviewers, we have performed additional experiments that substantially improve the manuscript's clarity and impact. Most notably, we conducted the suggested imaging experiments comparing RAW264.7 cells and bone marrow-derived macrophages (BMDMs) with fetal liver macrophages (FLMs). These experiments show that tentpoles are a prominent structure in RAW264.7 but exceedingly rare in BMDM and FLM (Supplementary Figure 3, Videos 15, 16) and are thus an epiphenomenon rather than a required structure for macropinocytosis. Moreover, these experiments illustrate that the morphologies observed during macropinocytosis are highly similar between BMDM and FLM. Additionally, we appreciate the recommended imaging experiments focused on the LY294002 treatment. These experiments illustrate that LY294002 only transiently interrupts ruffle formation, and it resumes within ~10 minutes with ruffle morphology similar to untreated cells as shown in new Supplementary Figure 2 and Video 7. We made textual and graphical improvements to the paper that are highlighted in cyan in the manuscript and detailed below. We enthusiastically resubmit our manuscript for publication in *Nature Communications*.

Editor Comments:

As you will see from the reports copied below, the reviewers raise important concerns. We find that these concerns limit the strength of the study, and therefore we ask you to address them with additional work. Without substantial revisions, we will be unlikely to send the paper back to review. We ask that you fully address all of the reviewer comments, with new experiments as appropriate, taking particular care to clarify experiments using LY294002 and to expand on PI3K localization.

If you feel that you are able to comprehensively address the reviewers' concerns, please provide a point-by-point response to these comments along with your revision. Please show all changes in the manuscript text file with track changes or colour highlighting. If you are unable to address specific reviewer requests or find any points invalid, please explain why in the point-by-point response.

Reviewer #1 (Remarks to the Author):

The uptake of fluid into macropinocytic vesicles is readily visible under the light microscope, but due to the scale and rapid movement of macropinocytic structures, it is difficult to describe in any detail by wide-field, confocal or spinning disc microscopy. Thus the morphologies that produce macropinosomes are not yet well established. Three have been suggested: cells project cups from their plasma membrane that contract at the lip and seal to release a macropinocytic vesicle into the cytoplasm (ref 13); or they form flaps that collapse back to the surface and fuse with it, trapping a droplet of medium; or finally they support and close cups using tentpoles (ref 12).

In contrast, lattice light sheet microscopy is perfect for the job. This paper presents lattice light sheet movies and images of stunning beauty using dual reporters for the plasma membrane and PIP3/PI3,4P2, to follow the formation of macropinosomes in macrophages. It shows that they often form from ruffles that collapse back onto the cell surface, or collide with each other, and then seal together. This mode occurs in un-stimulated macrophages and after stimulation with either CSF1 or LPS. These observations suggest that the spatial organization of macropinosome is less demanding than previously thought, and that the whole ruffle surface may be fusogenic.

PI3-kinase plays a central role in organizing macropinocytosis in both macrophages and *Dictyostelium amoebae*, but exactly how remains unknown. This work shows that macropinocytic ruffles are enriched in PIP3, giving general support to its importance, and that inhibition of PI3-kinase with LY294002 still allows ruffles to form, but prevents their sealing into macropinosomes, confirming previous work, but at better resolution.

In summary, this is an exciting paper, which helps us understand how macropinosomes form and the role of PIP3. The microscopy is beautiful. Publication would certainly advance the field. However, there are some issues that need addressing, as below.

1. It would be helpful if the authors could enquire further into why there are differing descriptions of how macropinosomes form (cups, tentpoles, ruffles colliding). Perhaps all three routes are true in

different circumstances or for different cells? Ideally, they should examine macrophages from the Swanson lab and RAW cells from the Stow lab in their conditions to test whether the differences are due to the different cell lines or conditions used previously.

We thank the reviewer for their very helpful comments. We have conducted additional LLSM imaging on RAW264.7 cells from the ATCC and primary BMDMs, both of which were expressing Mem-mNG, and the RAW264.7 were also expressing Lifeact-mSc, (Supplementary Figure 3, Videos 15, 16). These findings demonstrate that the RAW264.7 cells are unique (amongst the three cell lines tested) in their propensity to form filopodia on ruffles. The filopodial (tentpole) structures on RAW264.7 cells were observed with both fluorescent probes and appear to twist or collapse at sites of macropinocytosis consistent with the findings of Condon et al. (Condon et al., 2018). However, FLMs and BMDMs were highly similar in producing lamellar sheets and rarely displayed filopodial structures on ruffles or on forming macropinosomes, indicating that these structures are not required for macropinocytosis. We have added Supplementary Figure 3 and text clarifying these points (Lines 70-72, 238-244). We feel that these experiments and edits improve the clarity and impact of the paper's findings, and we appreciate the reviewer suggesting it.

2. Earlier work describes a stage before cup formation in which a linear ruffle circularizes and PIP3 accumulates within it (e.g. ref 22). Very similar images are produced here (eg movie 1, starting at 5:39), but now with the superior resolution of LLSM, are interpreted as being due to a ruffle folding back onto the cell surface and progressively fusing with it, followed by PIP3 accumulation on the sealed vesicle. Do the authors feel this is a reasonable interpretation, which might explain the earlier observations?

We agree with the reviewer's interpretation and have included a section in the discussion (Lines 245-255) elaborating on the superior resolution of the LLSM resulting in observation of PI3K activity throughout the ruffle as well as around the fully formed macropinosome. The circularization observed in phase-contrast microscopy (Yoshida, Hoppe, Araki, & Swanson, 2009) is consistent with the appearance of an ultimately spherical macropinosome; however, the higher 3D resolution of the LLSM revealed a much greater 3D complexity of membrane ruffles at the macropinosome formation site than could be observed by phase contrast.

And separately, do the authors see macropinosomes forming from cups? And if so what is their frequency?

We have not observed cup formations as seen in *Dictyostelium*. We have added to the discussion (Line 245-255) to further elaborate on the lack of cup-like formations and speculated on possible reasons for the differences.

3. The concentration of LY294002 used – 0.16 μM – is surprisingly low. The IC₅₀ for PI3-kinase activity is 1.4 μM and for fluid-phase uptake by macrophages, 3 μM (ref 24). How is it known that PI3-kinase is significantly inhibited, let alone so strongly that it can be stated that ruffling is independent of PIP3? What do higher doses do? A caveat to these experiments is that LY294002 inhibits both TORC1 and TORC2 (eg EMBO J 15, 5256, 1996).

We thank the reviewer for catching this mistake. We apologize for the error, and we have corrected the manuscript. The concentration of LY294002 used in these experiments was 50 μM (Araki, Egami, Watanabe, & Hatae, 2007). We have included additional work on the LY294002 treatment of macrophages as well (Video 7, Supplemental Figure 2) showing the rapid response to drug treatment by FLMs.

It should also be noted that the importance of PIP3 is strongly supported by *Dictyostelium* genetics, which shows that PI3-kinases are essential for macropinocytosis, with one of them having a specific role in closure (J Cell Sci 126, 4296 2013).

We have added text highlighting the previous work (especially the findings from *Dictyostelium*) (Lines 245-255) and thank you for bringing this to our attention. We are happy to acknowledge and utilize the significant work that has been done on macropinocytosis in *Dictyostelium*.

3. An idea advanced from *Dictyostelium* is that PIP3 domains activate actin polymerization at their edges by recruiting the WAVE complex there (ref 19). The formation of a circular ruffle around the PIP3 domain in giant circular ruffles shown in this paper supports the idea. Is this a new observation, or has it been reported before (I feel it has)?

We agree with the reviewer that these data are supportive of that idea. The complex circular dorsal ruffle seen in *Dictyostelium* is similar to the CSF-1 starvation/restimulation condition. This could be attributed to the circular dorsal ruffle being a macropinocytosis modality focused on survival/nutrient gathering. We have plans to investigate the recruitment of WAVE and WASP and their role in shaping the macropinosome.

The *Dictyostelium* model predicts that the extending edge of the ruffle, where actin polymerization occurs, should be a PIP3 domain boundary and therefore that one membrane of the ruffle should have PIP3 and the other not. This looks not to be true in many images, but is supported by a few. Could this be investigated in more detail, if the resolution and sectioning allow?

This is an interesting point and the imaging by (Hoeller et al., 2013) very clearly showed this. The probe used here likely is one reason for a more diffuse staining as it recognizes PIP3 and PI(3,4)P2. Additionally, the overall process of macropinocytosis is more variable compared to the highly regular formations in *Dictyostelium*. Notably, the macrophages have regions of their surface from which many ruffles extend and move laterally across their surface. This morphology appears distinct from the more focused cup-like structures observed in *Dictyostelium* and suggests that a larger patch of PIP3 membrane is present (or at least that there are many PIP3 positive ruffles within these regions). The strong accumulation of PIP3 that occurs during closure of the MP in the macrophage may indeed be the more equivalent structure to the PIP3 domain observed in *Dictyostelium*. We have added statements describing this idea to the Discussion (Lines 51-53, 245-255).

4. Minor things: please show scale bars and explain isosurface and volumetric intensity projections in a few simple words. How is the moment when a macropinosome seals identified? Please mark some sealing events in the movies. The cells look very flat - roughly how thick are they? And do they look like this in vivo?

We show scale bars when there was a true 2-dimensional image (i.e., SEM images, BF images, and 2D projections) and specified the view dimensions (x,y,z) in the legend for the 3D data which is shown as an outline box. This is caused by the 3-dimensional perspective, such that 5 microns is different when referring to the front of the volume versus the back of the volume. The bounding box takes this into consideration.

To improve the clarity on how macropinosome sealing was identified, we added a section to the results (Lines 115-120), and we have added a clearer description of the different visualization terms and how they function in the results section (Lines 85-92, 9100).

The cells are roughly 6 microns tall in the center but are very thin along the edges (~ 1 micron) (Lines 115-117). We have not imaged these cells in vivo; however, we have included the comparison with BMDM that showed similar characteristics (Lines 69-71, 238-244, Supplementary Figure 3, Video 15).

Reviewer #2 (Remarks to the Author):

The work presented by Dr. Scott and colleagues entitled "The structural dynamics of macropinosome formation and PI3-kinase-mediated sealing revealed by lattice lightsheet microscopy", which shows the biological findings presumably based on the live imaging analysis from LLSM featuring with higher axial resolution. Taking the advantages of the high spatiotemporal resolution and low sample damage from LLSM, they revisit the micropinocytosis in macrophage-like cell lines and shows that PI3K activity is amplified in ruffles and critical for sealing them into macropinosomes, less effect on the ruffling rate and morphology. They did a lot of the live imaging (122 macropinosome formations) and got the conclusions based on the visualization techniques from the 4D (xyz +t) data with and without drug treatment. A lot of efforts have been put for collecting the live imaging data in order to conclude their new findings, which is a hard work due to the complicated management for live imaging. The reviewer

appreciates their job for multicolor 4D imaging analysis. Through the manuscript, they used a lot of descriptions to explain the image data along with time frame. Given that more quantitative analysis with time evolving, that will merit these 5D imaging data (x, y, z, t, λ). Overall, the manuscript is well-written and the figures support their findings, which is good for the publication. However, the reviewer would like to ask the authors to address the following concerns,

1. In the introduction part, line50-53, the authors make the arguments about the PI3K activity is important for macropinosome formation. But in the experimental results, they monitored AktPH, probing the products of PI3K. Although the AktPH seems has been used to characterize PI3K activity during micropinocytosis, the reviewer wonders whether direct watching the PI3K activity is feasible or not, that is probing PI3K. Moreover, the reviewer didn't see any immediately increase or decrease the PI3K activity or PI3P/PI2P methods in your manuscript. For example, the Rapamycin inducible dimerization system to increase or decrease the PI3P/PI2P.

Thank you for these comments. We have revised the text to more clearly indicate that the AktPH probe binds to PIP3 and PI(3,4)P2 (Lines 84-85). PIP3 is the direct product of PI3K, and PI(3,4)P2 is generated by action of SHIP1/2. This has been used previously in the literature as a proxy for PI3K activity (Yoshida et al., 2015). Originally, we inhibited PI3K with LY294002 for 30 min and showed a clearance of AktPH from the plasma membrane in Figure 3e vs Figure 3b. We have created additional material (Supplementary Figure 2, Movie 7) to further emphasize the LY294002 inhibition in hopes of clarifying this. The changes in PIP3 intensity are most striking during and immediately following closure of the macropinosome (Fig. 1c,d,e Fig. 2a Fig. 4a,c). However, thanks to the suggested additional LY294002 work, the side-by-side comparison of the same cell before and after treatment displays the intensity difference much better (Supplementary Figure 2).

2. For all the figures and imaging analysis, the processed or imaging rendering have been applied to the presented data, the reviewer still thinks that the raw data or live images should be given at least for one data set or presented figures. The rendering software ChimeraX, there are many parameters involved for the imaging rendering. Therefore, raw imaging should be given for clarification. Moreover, there is an intensity plot along the image such as in figure 2b, the review think this should be done in raw image instead of processed one.

We completely agree that there are many parameters for image rendering, and it takes time and effort to do it properly. We have added some text to the deconvolution and post processing Methods section to add clarity (Lines 364-366, 381-383). Given the nature of dual objective lightsheet instruments (Gao, Shao, Chen, & Betzig, 2014) the non-deskewed raw data is of little value for displaying or quantifying intensities. For visualization, we ensured that the intensity vs opacity was linear for the projections and used the same imaging conditions and display settings to make comparison of the intensities possible. This is a difference from the default settings in most commercial packages which use a gamma curve that further complicates the analysis. In the intensity plot, we were focused on the overall trend in terms of the spatial correlation, and not the absolute intensities of the images. If this were the case, then we absolutely would have used the deskewed, but non-deconvolved data.

3. In line114, the corresponding area of the macropinosome process also need to be marked in isosurface view in Figure 2a. In Figure 2b, the intensity plot is made from the 3D maximum intensity projection? Since this a 3D data, not sure this is a good way to present this? Since this is a time-evolving process, an intensity plot with "time" variable for the description for different levels of PI3K activity would probably be a better way.

We have added an arrow to the isosurface images of Figure 2, indicating the macropinosomes, and updated the legend; our apologies for omitting that. The reviewer touches on a very important point, and one that warrants a high-impact paper of its own and that is the quantification of volumetric data over time. This is an area of intense research, and it has not been realized yet, as one needs to reduce the 4D data down to 2D (intensity vs time) while somehow maintaining the 3D nature of the intensity. We attempt to maintain the volumetric nature and allow for direct comparison between conditions by using split fluorescent channels with identical imaging, processing, and visualization settings (Fig 3b vs Fig 3e) to make the comparison of relative AktPH in the ruffle volume in control and LY294002-treated cells (Supplementary Figure 2). The maximum intensity projection (MIP) is used, as we are not

correlating the 3D space with the intensity, but rather the variations between formation peak levels and ruffle levels.

4. In Fig. 3, the PI3K inhibitor LY294002 shows the different levels of effect on the macropinosomes formation. Presumably PI3K activity affects the sealing process. Although the isosurface view gives the good insight for the macropinosome formation, the presentation lacks the intensity information, especially for PI3K. If possible, the reviewer would like to see how the PI3K intensity changes during the macropinosome formation with time axis within the whole macropinosome by volume displaying with intensity colormap. That will make all these data more valuable. A related supplementary movie shows the intensity change of macropinosome will be better.

The intensity of Scarlet-AktPH localization to ruffles was dramatically diminished by LY294002. We present the data side-by-side and co-scaled with the untreated control (Fig. 3). We have updated the text and added additional supplemental movies showing the clearance of the Scarlet-AktPH from the plasma membrane upon LY treatment (Lines 145-154, Supplementary Figure 2, Video 7). We believe that this should make interpretation of these images clearer.

5. For drug treatment part, how long does PI3K inhibitor LY294002 take to affect the cells? According to the experimental parts in line 246, 30 mins pre-incubation is used. Could you do the live-cell imaging which including before and after drug dosing? That would solidify the idea about "PI3K activity primes ruffles for fusion to seal nascent macropinosomes". And if the removal of the drugs, do the cells recover from the effect? What's difference between CSF-1 and LPS in your system? Can you use same analysis tool (figure 7e) to analyze the CSF-1 treated cells?

Thank you for this suggestion. We conducted this experiment and added Supplemental Figure 2 that shows pre- and 1 min following treatment as well as a corresponding video (Video 7). As can be observed here, LY294002 leads to a clearance of the AktPH from the plasma membrane and a transient arrest of ruffling. After treatment, ruffling resumes similar to the 30-min treatment of our other work and still no macropinosomes form. We have revised the text to point this out and strengthen the point that PI3K activity primes ruffles for fusion (Lines 149-155).

We included both CSF-1 and LPS to align with other studies examining stimulated macropinocytosis and to observe differences in their potential to stimulate different macropinocytic morphologies (Condon et al., 2018). We have updated the text to more clearly point out that while some differences in ruffle structure were observed, we did not see tentpole formation in FLM cells (Lines 69-71, 238-255).

6. For temporal resolution, while the review went through the movies, it seems that the membrane ruffling and macropinosomes formation have different rate. The initial step is slower and macropinosome is faster. It seems the equal time interval is used for the live imaging resulting in lacking details for the macropinosome formation. For example, supplementary figure 1 shows the constitutive micropinocytosis. The complicated structures formed at the lateral time points. If the better temporal resolution is applied, the fine details will be obtained for the macropinosome formation.

We also think that faster is better; unfortunately, there are limits to the speed that our current instrument can obtain data with sufficient signal:noise ratio, while also maintaining a minimal amount of photobleaching. Unfortunately, we are at that current limit for our instrument when using our conditions. With that said, the formation of membrane extension and macropinosomes appears relatively smooth when reviewed frame by frame, i.e., there are not large jumps of the membrane between frames so from a sampling standpoint with respect to the claims made in the manuscript we believe we have sufficiently sampled the behavior.

We appreciate the reviewer working through all of the movies and hope they understand we spent a great deal of time selecting a limited number of frames, a highly restricted field-of-view, and the most representative cells to fit in extremely limited 2D figure format of published literature while not losing the core statement of each figure. In many cases the videos become clearer with each viewing and in many cases simply playing through them frame by frame clarifies the activity. Unfortunately, a 2D figure will always fail to retain the complex 4D information and we hope that the videos supplement that

limitation.

7. For spatial resolution, in figure 5, there is a nice image for SEM, but no scale bar information. The reviewer would like to see the thickness of ruffling membrane could be measured or not? That probably gives you the sense about your optical microscope image vs high resolution SEM image. Especially, your 3D surface rendering way will broad your spatial resolution.

We thank the reviewer for pointing this out and have included a description of the thickness of our cells (Lines 114-116) as well as added the scale bar in Fig 5b.

8. The other concern is about the membrane density vs PI3K density. In the figures, the PI3K shows the high intensity at the base of macropinosome, is it possible due to the accumulation of membrane.

Thank you for this comment. We closely examined the concern of membrane accumulation as the cause of the PI3K intensity. In Figures 2 and 4, we show the volumetric intensity of both the membrane (green) and AktPH (magenta). From these figures we conclude that AktPH accumulates at these sites due to increased density of PI3K products and not due to a large increase in membrane.

9. Since the work present the time evolving data for micropinocytosis event, again the reviewer hopes to see the 3D time profiles of intensity and morphology for the event and by doing this will merit the high spatiotemporal resolution of LLSM.

At the current state of volumetric computation there is a limit to the possible ways to intuitively reduce the number of dimensions to show a 2D graph that maintains the 3D intensity values over time. We put a significant amount of effort and time into properly visualizing the data. We attempted to provide all the elements to the reader including intensity, x-y-z spatial structure, multiple fluorescent channels, and signal evolution over time. For example, in Figure 5, we dedicated a section to the structure of the membrane (Panel c), the relative activity of PI3K (Panel d), and we made an overlay of the structure and volume (Panel e) to highlight where exactly the increased localization of AktPH was with respect to the extended ruffles. Finally, we provided videos (Video 8,9) to combine the information to ensure the reader has access to as much of the data as we can show.

Araki, N., Egami, Y., Watanabe, Y., & Hatae, T. (2007). Phosphoinositide metabolism during membrane ruffling and macropinosome formation in EGF-stimulated A431 cells. *Exp Cell Res*, 313(7), 1496-1507. doi:10.1016/j.yexcr.2007.02.012

Condon, N. D., Heddleston, J. M., Chew, T.-L., Luo, L., McPherson, P. S., Ioannou, M. S., . . . Wall, A. A. (2018). Macropinosome formation by tent pole ruffling in macrophages. *Journal of Cell Biology*, 217(11), 3873-3885. doi:10.1083/jcb.201804137

Gao, L., Shao, L., Chen, B.-C., & Betzig, E. (2014). 3D live fluorescence imaging of cellular dynamics using Bessel beam plane illumination microscopy. *Nature Protocols*, 9(5), 1083-1101. doi:10.1038/nprot.2014.087

Hoeller, O., Bolourani, P., Clark, J., Stephens, L. R., Hawkins, P. T., Weiner, O. D., . . . Kay, R. R. (2013). Two distinct functions for PI3-kinases in macropinocytosis. *Journal of Cell Science*, 126(18), 4296. doi:10.1242/jcs.134015

Yoshida, S., Gaeta, I., Pacitto, R., Krienke, L., Alge, O., Gregorka, B., & Swanson, J. A. (2015). Differential signaling during macropinocytosis in response to M-CSF and PMA in macrophages. *Frontiers in physiology*, 6, 8-8. doi:10.3389/fphys.2015.00008

Yoshida, S., Hoppe, A. D., Araki, N., & Swanson, J. A. (2009). Sequential signaling in plasma-membrane domains during macropinosome formation in macrophages. *Journal of Cell Science*, 122(18), 3250-3261. doi:10.1242/jcs.053207

REVIEWERS' COMMENTS

Reviewer #1 (Remarks to the Author):

This paper has been substantially improved. It is very good to see the comparison of macropinocytosis between 3 different cell lines, the further experiments with the LY drug and the improved contextualization of the work with the literature.

I am still surprised at the lack of macropinosomes forming from cup-like structures, and wonder if there is some underlying reason? However, the results seem clear and establish that there are routes to macropinosome formation by membrane collision. Doubtless there will be much more work in the future with additional reporters, cell lines and conditions to throw further light on the issue

Reviewer #2 (Remarks to the Author):

Thanks for the authors addressed a lot of details regarding the raised questions. The review appreciated additional experiments for the improvement of the paper quality. As the authors mentioned in the reply letter, As for the 5D (x,y,z,t,λ) image analysis, indeed it is very hard to present the data in a very right way. I am impressed by all the data presented herein are all live imaging data except SEM, and fit the scope of the story they would like to tell. I am satisfied with the revised manuscript without further questions.

CBC

We are very appreciative of the reviewers' efforts that has resulted in a much-improved manuscript.

Reviewer #1 (Remarks to the Author):

This paper has been substantially improved. It is very good to see the comparison of macropinocytosis between 3 different cell lines, the further experiments with the LY drug and the improved contextualization of the work with the literature.

I am still surprised at the lack of macropinosomes forming from cup-like structures, and wonder if there is some underlying reason? However, the results seem clear and establish that there are routes to macropinosome formation by membrane collision. Doubtless there will be much more work in the future with additional reporters, cell lines and conditions to throw further light on the issue

Reviewer #2 (Remarks to the Author):

Thanks for the authors addressed a lot of details regarding the raised questions. The review appreciated additional experiments for the improvement of the paper quality. As the authors mentioned in the reply letter, As for the 5D (x,y,z,t,λ) image analysis, indeed it is very hard to present the data in a very right way. I am impressed by all the data presented herein are all live imaging data except SEM, and fit the scope of the story they would like to tell. I am satisfied with the revised manuscript without further questions.

CBC